# Integrating Enhanced Profiling and Chemometrics to Unveil the Potential Markers for Differentiating among the Leaves of *Panax ginseng*, *P. quinquefolius,* and *P. notoginseng* by Ultra-High Performance Liquid Chromatography/Ion Mobility-Quadrupole Time-of-Flight Mass Spectrometry

**DOI:** 10.3390/molecules27175549

**Published:** 2022-08-29

**Authors:** Feifei Yang, Boxue Chen, Meiting Jiang, Huimin Wang, Ying Hu, Hongda Wang, Xiaoyan Xu, Xiumei Gao, Wenzhi Yang

**Affiliations:** 1State Key Laboratory of Component-based Chinese Medicine, Tianjin University of Traditional Chinese Medicine, 10 Poyanghu Road, Jinghai, Tianjin 301617, China; 2Haihe Laboratory of Modern Chinese Medicine, Tianjin University of Traditional Chinese Medicine, 10 Poyanghu Road, Jinghai, Tianjin 301617, China; 3Key Laboratory of Pharmacology of Traditional Chinese Medical Formulae, Ministry of Education, Tianjin University of Traditional Chinese Medicine, 10 Poyanghu Road, Jinghai, Tianjin 301617, China

**Keywords:** *panax*, leaf, UHPLC/IM-QTOF-MS, metabolomics, ginsenoside, chemical marker

## Abstract

The leaves of *Panax* species (e.g., *Panax ginseng*-PGL, *P. quinquefolius*-PQL, and *P. notoginseng*-PNL) can serve as a source for healthcare products. Comprehensive characterization and unveiling of the metabolomic difference among PGL, PQL, and PNL are critical to ensure their correct use. For this purpose, enhanced profiling and chemometrics were integrated to probe into the ginsenoside markers for PGL/PQL/PNL by ultra-high performance liquid chromatography/ion mobility-quadrupole time-of-flight mass spectrometry (UHPLC/IM-QTOF-MS). A hybrid scan approach (HDMS^E^-HDDDA) was established achieving the dimension-enhanced metabolic profiling, with 342 saponins identified or tentatively characterized from PGL/PQL/PNL. Multivariate statistical analysis (33 batches of leaf samples) could unveil 42 marker saponins, and the characteristic ginsenosides diagnostic for differentiating among PGL/PQL/PNL were primarily established. Compared with the single DDA or DIA, the HDMS^E^-HDDDA hybrid scan approach could balance between the metabolome coverage and spectral reliability, leading to high-definition MS spectra and the additional collision-cross section (CCS) useful to differentiate isomers.

## 1. Introduction

The *Panax* genus serves as the source for a great variety of natural products showing beneficial effects to humans. Diverse formulation products consisting of the extracts of the *Panax* herbs are popularly consumed on a global scope, such as functional foods, toothpaste, cigarettes, soaps, cosmetics, beverages (including beer), coffee, baby foods, candies, and gum, in addition to the extensive clinical use [1,2,3]. Multiple *Panax* species have exhibited an extensive range of pharmacological effects on cognition and blood circulation as well as antitumor, antioxidative, and antifatigue effects on human health [4,5,6], of which *P. ginseng* (Asian or Korean ginseng), *P. quinquefolius* L. (American ginseng), *P. notoginseng* (Burk.) F.H. Chen (Sanchi ginseng) are the three most well-known [7,8]. Diverse classes of herbal metabolites, such as the polysaccharides, saponins, flavonoids, organic acids (esters), amino acids, sterols, carbenes, *etc.*, have been reported from more than ten different *Panax* species [9,10]. The contained saponins, also known as ginsenosides, are the main substances related to the pharmacological effects of ginseng, which also act as index components for the purpose of quality control of different products [11]. Until now, a collection of documents is available regarding the metabolic difference studies of the same parts (roots [12,13,14] or flowers [15,16]) among different *Panax* species, the different parts of the *Panax* species [17], and their combination [18], which can demonstrate that ginsenosides have great potential to differentiate ginseng products. In contrast to the roots covering very extensive studies, the leaves of the three most important *Panax* species (leaf of *P. ginseng*, PGL; leaf of *P. quinquefolius*, PQL; leaf of *P. notoginseng*, PNL) have been rarely investigated, in particular regarding the differences in their metabolome. To characterize the chemical components and to discern the botanical source are two key issues to ensure the correct use of ginseng products. Currently available approaches dedicated to the systematic multi-component characterization from an herb or a biosample mostly rely on liquid chromatography-mass spectrometry (LC-MS), which has shown high selectivity (the flexible use of reversed-phase chromatography [19], hydrophilic interaction chromatography [20], and ion-exchange chromatography [21], *etc.*), high sensitivity (diverse fit-for-purpose MS experiments), and high universality (covering most of the known classes of chemical structures). Moreover, LC-MS profiling combined with chemometrics can render the untargeted metabolomics analysis suitable for discovering potential chemical markers for the authentication and quality evaluation of the easily confused herb species in a holistic manner [22].

Untargeted profiling strategies by LC-MS are preferably employed to profile the composition of medicinal herbs without the need for any pre-knowledge, which mainly relies on the use of data-dependent acquisition (DDA) or data-independent acquisition (DIA). DDA strategy is more easily accessible, which acquires the MS/MS or MS^n^ data for a subset of the total ions according to some criteria, such as the intensity, isotopic pattern, neutral loss, diagnostic fragment ions, and included or excluded *m/z*, by collision-induced dissociation (CID) or high-energy collision-induced dissociation (HCD) [23]. The resultant DDA data are definite and comparatively easy to interpret, but its coverage of the components of interest may be restricted when facing very complex chemical matrices or complicated precursor ion species (composed of the diverse adducts, dimers, in-source decay fragments, and even doubly charged ions, *etc.*) [24]. To address this issue, numerous efforts have been made. The inclusion of a precursor ion list (PIL) containing the targeted structure information in DDA can significantly improve the sensitivity in characterizing those co-eluting minor metabolites, as the scanned targeted precursors are endowed with the highest priority to acquire the MS^n^ data. Moreover, the utilization of an exclusion list (EL) in DDA is also beneficial to increase the coverage of DDA [25,26]. Notably, data-independent acquisition (DIA) has currently attracted more interest, by which the MS^2^ data of all precursor ions within the predefined scan range (MS^1^) can be performed without the omission or difference. DIA typically provides a broader dynamic range, higher ginsenosides identification rates, improved reproducibility of identification, and accuracy for quantification experiments [27]. Commonly employed DIA approaches, including all ions fragmentation (AIF) [28,29] and MS^All^ (also called MS^E^ [29,30]) that perform once fragmentation of all the scanned precursors and sequential window acquisition of all theoretical spectra (SWATH) has occurred [29]. Despite no MS^2^ information missing by DIA, precursor-product ion matching is necessary prior to the MS^2^ data interpretation [31]. Additionally, the introduction of ion mobility mass spectrometry (IM-MS) provides an additional dimension of separation (based on the size, shape, and charge state of the gas-phase ions) that is orthogonal to MS and gives collision cross section (CCS) information. Notably, the CCS shows great potential in discriminating metabolite isomers [32,33,34]. On the other hand, an untargeted metabolomics platform can offer useful insights into the evaluation of the quality of traditional Chinese medicines, including easily confusing species [35,36], species of different geographic origins [37], different parts of the same plant [38], and different growing ages amongst species [39,40]. Pseudo-targeted metabolomics has been proven to be feasible with a variety of advantages, including the improvement in the detection sensitivity for those low-abundance metabolites, which can thus expand the screening range of the potential biomarkers in metabolomics studies [41,42,43].

The purpose of this work was to probe into the metabolic difference among PGL, PQL, and PNL, and primarily establish the “Identification Markers” enabling their differentiation. By a feat of the advanced Vion™ IM-QTOF high-resolution LC-MS platform, a potent hybrid scan approach (HDMS^E^-HDDDA), enabling the IM separation of all precursors and the alternate DIA/DDA acquisitions in once injection analysis, was developed to record the CID-MS^2^ data in the negative mode for ginsenosides characterization [12,44], and the untargeted metabolomics differential technique was utilized to discover ginsenoside markers diagnostic for differentiating PGL/PQL/PNL, by analyzing thirty-three batches of leaf samples (Appendix A). The overall technical roadmap is displayed in Figure 1. Particularly, to boost the coverage of HDDDA in the hybrid scan approach, a PIL consisting of 174 known ginsenoside masses was included. Moreover, a total of 60 ginsenosides were used as the reference compounds (Appendix A). We could achieve the comprehensive ginsenosides identification simultaneously from the leaf samples of three *Panax* species (PGL/PQL/PNL) and unveil the ginsenoside markers with potential for their differentiation. The merits of the HDMS^E^-HDDDA hybrid scan over the single DIA or DDA approach were discussed, as well.

## 2. Results and Discussion

### 2.1. Optimization of a UHPLC/IM-QTOF-MS Approach for the Enhanced Profiling and Characterization of Ginsenosides Simultaneously from PGL, PQL, and PNL

Due to the chemical complexity of herb-derived natural products, a powerful analytical strategy is greatly needed to elucidate the contained components. In this work, on the Vion IM-QTOF platform, we aimed to develop a hybrid scan approach (HDMS^E^-HDDDA), in combination with the automatic peak annotation workflows of UNIFI, to comprehensively characterize ginsenosides from PGL/PQL/PNL. To maximize the resolution and sensitivity of detection, key parameters both in UHPLC separation and MS detection were optimized in sequence by single-factor experiments using a QC sample (QC1). In our previous research [15,38], it was found that different additives in the water phase of the mobile phase (such as ammonium acetate and formic acid) could remarkably influence the retention behavior of ginsenosides on an RP-mode HSS T3 column. The addition of 0.1% formic acid in the water phase could separate more ginsenoside peaks. The stationary phase is a key factor affecting the resolution of the analytes, and here we evaluated the separation performance of ginsenosides on ten UHPLC columns (with different packing materials and bonding technologies), by comparing the number of resolvable peaks and the peak symmetry. Ten sub-2 µm particles packed columns from two suppliers (Waters and Agilent), including HSS T3, BEH Shield RP18, CORTECS UPLC C18+, CSH C18, HSS C18 SB, BEH C18 (Waters, Milford, MA, USA), ZORBAX SB-C18, ZORBAX Eclipse Plus C18, ZORBAX Extend C18, and ZORBAX SB-Aq (Agilent Technologies, Waldbronn, Germany) were used (Figure 2). According to the results, HSS T3 and BEH Shield RP18 could enable good separation and peak shape for ginsenosides. By further comparing the resolvable peaks under the same elution gradient, BEH Shield RP18 could be a better choice, as it allowed higher analysis efficiency and more elaborate separation around seven minutes for a pair of ginsenoside markers, Rg1/Re, compared with those on the HSS T3 (9466) column. We thus selected the BEH Shield RP18 column for subsequent experiments. Moreover, the influence of column temperature (from 25 to 40 °C) on the BEH RP18 column could exert a very minor effect on the resolution performance. Comparatively, 30 °C was considered appropriate (Appendix A). The gradient elution program was further adjusted, and as a result, satisfactory resolution of ginsenosides simultaneously from PGL, PQL, and PNL, was achieved within 20 min.

The parameters involved in the HDMS^E^-HDDDA hybrid scan approach were optimized in the next step. Our previous work obtained the best settings of spray voltage (1.0 kV) and cone voltage (20 V) for monitoring ginsenosides in the negative ESI mode [15]. The hybrid scan approach of HDMS^E^-HDDDA was composed of two collision energy settings, that is, RCE in HDMS^E^ [17] and MDRCE in HDDDA [45]. We separately optimized RCE and MDRCE to gain high-quality of MS^2^ spectra for ginsenosides. Consequently, the higher levels of collision energy ramps benefitted the elimination of sugars from the deprotonated ginsenoside molecules and the generation of free sapogenin ions. For HDMS^E^, six levels of RCE, including 20–40 eV, 40–60 eV, 60–80 eV, 80–100 eV, 80–120 eV, and 120–140 eV were compared using the QC1 sample by observing the richness of the sugar lost fragments and the sapogenin-related product ions for ginsenosides Rb2, Re, pseudo-F11, and m-Rb2. We intended to select the ramp of 80–100 eV as the collision energy in HDMS^E^, as all these different index saponins generated diversified product ions (Appendix A). Similarly, six levels of MDRCE, involving Low mass and High mass: 20–40/40–60 eV, 30–50/50–70 eV, 40–60/60–80 eV, 50–70/70–90 eV, 20–60/40–90 eV, and 20–70/70–100 eV, were compared by observing the fragments in the MS^2^ spectra for ginsenosides Rb2, Rd, p-F11, and m-Rd using the QC1 sample. Evidently, the MDRCE of 30–50/50–70 eV could produce diversified MS^2^ fragments for most of the index saponins benefitting their structural elucidation (Appendix A) [17].

### 2.2. Superiority Demonstration for the HDMS^E^-PIL/HDDDA Hybrid Scan Approach

The Vion IM-QTOF LC-MS platform can enable diverse MS experiments, such as MS/HDMS, MS^E^/HDMS^E^, DDA/HDDDA, and the hybrid scan approach of HDMS^E^-MS/MS (HDMS^E^-HDDDA). Considering that the inclusion of PIL in DDA was able to greatly expand the coverage of the components of interest [19,21,44], in the current work, we evaluated the performance of four MS^2^ acquisition approaches, involving the HDMS^E^, DDA, DDA-PIL, and the hybrid HDMS^E^-PIL/HDDDA in profiling the ginsenosides from the mixed QC1 sample. Evidently, the superiority of the established hybrid scan approach could be reflected in the following three aspects: (1) the coverage of targeted mass; (2) more resolvable and more reliable mass spectra; and (3) the provision of CCS information useful for discriminating the isomers.

Firstly, the PIL-HDDDA within the hybrid scan method showed better performance in the numbers of the “Identified Components” (263; the components automatedly identified by UNIFI using the database) than DDA (77) and DDA-PIL (194) and HDMS^E^ (Figure 3A). However, a single HDMS^E^ could lead to more primarily identified components than the hybrid approach (835 VS. 559). Moreover, in the case of the ″Unknown Components″ (the components failing to match with the library by UNIFI) obtained by either HDMS^E^ or HDDDA, the hybrid scan approach gave no compromise to the single HDMS^E^ which acquired the largest amount of MS^2^ spectra; some within this list could be manually analyzed to complement the characterized components. Secondly, compared with DDA and DDA-PIL without IM separation, the obtained MS^1^ spectra were clearer, with some interference ions removed (Figure 3B), using ginsenoside Ra1 as a case. In addition, by comparing the MS^2^ spectra obtained by HDMS^E^ and HDDDA within the hybrid scan approach, the one by HDDDA suffered from much less interference than that by HDMS^E^ (using ginsenoside Rb3), testifying the high spectral quality of DDA. As illustrated in the heat map, six groups of co-eluting components at 5.45, 8.40, 10.08, 12.55, 12.98, and 17.64 min, and six groups of isomers detected at *m/z* 1031.5445, 1139.5858, 961.5368, 799.4829, 1163.5855, and 679.4426 could be resolved due to the enabling of IM separation (Figure 3C,D). Thirdly, the hybrid scan approach could provide the CCS information, which displayed the potential for discriminating isomers [19,44,46]. Figure 3E illustrated the CCS distribution and difference determined for six groups of isomers. High positive correlation between the determined *m/z* and CCS values was observed for the ginsenosides characterized from PGL/PQL/PNL, showing the linearity correlation coefficient *R^2^* at 0.8067 (Figure 3F), which could be useful to discern false positive identifications generated by the software. Taken together, this hybrid scan approach (HDMS^E^-PIL/HDDDA) could represent a powerful strategy to support comprehensive untargeted metabolites characterization.

### 2.3. In-Depth Characterization of Ginsenosides from PGL, PQL, and PNL Based on the HDMS^E^-PIL/HDDDA CID-MS^2^ Data and UNIFI Intelligent Peak Annotation Workflows

By use of the powerful hybrid scan approach and well-established intelligent peak annotation workflows, the negative CID-MS^2^ data were interpreted to comprehensively characterize the ginsenosides simultaneously from the leaves of three *Panax* species (PGL/PQL/PNL). After efficient and separate processing of the HDMS^E^ and HDDDA MS^2^ data, those listed as “Identified Compounds” were further confirmed based on the *t*_R_, fragmentation pathways analysis, precursor-ion adduct forms, and the elution sequence on RP column, whilst those involved in “Unknown Compounds” were manually identified by the aforementioned rules [19]. By these efforts, we could identify or tentatively characterize 342 saponins (Appendix A), including 199 from PGL, 150 from PQL, and 158 from PNL, respectively. Amongst them, 45 compounds were identified by comparison with the reference compounds (including *t*_R_, MS^1^, and MS^2^). Figure 4 shows the base peak chromatograms of representative samples of PGL, PQL, and PNL by annotating those peaks identified with the aid of reference standards. In addition, after further searching against the in-house database consisting of 573 ginsenosides, 81 ginsenosides may have not been isolated from the whole *Panax* genus. In structure, the differences in the sapogenins and the acyl substituents, these characterized ginsenosides could be divided into the protopanaxatriol (PPT) type, protopanaxadiol (PPD) type, oleanolic acid (OA) type, ocotillol (OT) type, C-17 side-chain varied, the malonylated, and the others [33,47]. The neutral loss (NL) masses of 162.0528 Da, 146.0579 Da, 132.0423 Da, and 176.0321 Da, could be consistent with the eliminations of Glc (glucose), Rha (rhamnose), a pentose (Xylose-Xyl, and Arabinose-Ara in either the pyran- or furan-form), and GlurA (glucuronic acid), respectively [17]. For the convenient depiction, Xyl was used to express all the pentose characterized by NL of 132.0423 Da for the ginsenosides characterized from PGL/PQL/PNL (Appendix A).

Ginsenosides of the PPT and PPD types can represent two very important subclasses of tetracyclic saponins occurring in most of the *Panax* species [9,11]. A total of 69 PPT-type (accounting for 20.18%) and 92 PPD-type (26.90%) ginsenosides were characterized from PGL, PQL, and PNL, in the current work. The sapogenin fragments at *m/z* 475.3787 ([PPT−H]^−^)/391.2848 ([PPT−H−C_6_H_12_]^−^) and *m/z* 459.3838 ([PPD−H]^−^)/375.2899 ([PPD−H−C_6_H_12_]^−^) could be used as the diagnostic product ions (DPIs) to rapidly characterize the PPT-type and PPD-type saponins, respectively [47]. Furthermore, the accompanying formic acid adduct and the genuine deprotonated precursor were remarkable features in the MS^1^ spectra for both the PPT- and PPD-type ginsenosides [14]. The ready eliminations of sugars and the generation of characteristic sapogenin ions were the major fragmentation features for them, which could be illustrated by a PPD-type reference compound, ginsenoside Rb2 (Figure 5A). Its deprotonated precursor ion was detected at *m/z* 1077.5851 (*t*_R_ 12.27 min), accompanied by the FA-adduct ion at *m/z* 1123.5900, which could inform the molecular formula of C_53_H_90_O_22_. It was dissociated into the fragments at *m/z* 945.5431, 783.4901, 621.4370, 459.3839, and 375.2903, which were consistent with the successive NL of one Xyl (132.0423 Da) and three Glc groups (3 × 162.0528 Da) from the deprotonated precursor ions. Similarly, in the case of an unknown compound **244**# (*t*_R_ 15.35 min), the rich precursor ions were observed at *m/z* 987.5537 ([M−H]^−^). In its MS^2^ spectrum, the product ion at *m/z* 945.5435 was assigned as [M−H−Ace]^−^, and the presence of three Glc residues could be deduced based on the successive NL of 3 × 162.0528 Da (*m/z* 945.5435 > 783.4873 > 621.4395 > 459.3827; Figure 5A). By comparing with the in-house ginsenoside library, we could primarily characterize this compound as 6″-*O*-acetylginsenoside Rg3 or isomer (PPD-3Glc-Ace).

Ginsenosides of the PPT and PPD types can represent two very important subclasses of tetracyclic saponins occurring to most of the *Panax* species [9,11]. A total of 69 PPT-type (accounting for 20.18%) and 92 PPD-type (26.90%) ginsenosides were characterized from PGL, PQL, and PNL, in the current work. The sapogenin fragments at *m/z* 475.3787 ([PPT−H]^−^)/391.2848 ([PPT−H−C_6_H_12_]^−^) and *m/z* 459.3838 ([PPD−H]^−^)/375.2899 ([PPD−H−C_6_H_12_]^−^) could be used as the diagnostic product ions (DPIs) to rapidly characterize the PPT-type and PPD-type saponins, respectively [47]. What is more, the accompanied formic acid adduct and the genuine deprotonated precursor were the remarkable features in the MS^1^ spectra for both the PPT- and PPD-type ginsenosides [14]. The ready eliminations of sugars and the generation of characteristic sapogenin ions were the major fragmentation features for them, which could be illustrated by a PPD-type reference compound, ginsenoside Rb2 (Figure 5A). Its deprotonated precursor ion was detected at *m/z* 1077.5834 (*t*_R_ 12.27 min), accompanied by the FA-adduct ion at *m/z* 1123.5900, which could inform the molecular formula of C_53_H_90_O_22_. It was dissociated into the fragments at *m/z* 945.5396, 783.4885, 621.4357, and 459.3852, which were consistent with the successive NL of one Xyl (132.0423 Da) and three Glc groups (3 × 162.0528 Da) from the deprotonated precursor ions. Similarly, in the case of an unknown compound **244**# (*t*_R_ 15.35 min), the rich precursor ions were observed at *m/z* 987.5537 ([M−H]^−^). In its MS^2^ spectrum, the product ion at *m*/*z* 945.5435 was assigned as [M−H−Ace]^−^, and the presence of three Glc residues could be deduced based on the successive NL of 3 × 162.0528 Da (*m*/*z* 945.5435 > 783.4873 > 621.4395 > 459.3827; Figure 5A). By comparing with the in-house ginsenoside library, we could primarily characterize this compound as 6″-*O*-acetylginsenoside Rg3 or isomer (PPD-3Glc-Ace).

Malonylginsenosides are a class of acidic saponins containing free carboxyl, which are widely distributed in various *Panax* species, in particular in the flower parts [15,48]. In this work, we could characterize 42 malonyl ginsenosides (12.28% of the total), and five thereof were identified by comparison with the reference compounds (**79**#/**159**#/**200**#/**246**#/**254**#). Compared with the neutral ginsenosides, malonylginsenosides could be easily discerned based on their rich deprotonated precursors together with the in-source CO_2_-eliminated fragments by full scan [13,48]. Moreover, the negative CID-MS^2^ fragmentation features of malonylginsenosides involved the neutral elimination of CO_2_ (43.9898 Da) and a whole malonyl group (86.0004 Da) (C_3_H_2_O_3_) or plus a molecule of H_2_O (104.0110 Da) or even the dimalonyl groups (172.0008 Da) (2 × C_3_H_2_O_3_) or plus a molecule of H_2_O (190.0114 Da), in addition to the common NL of sugars and the generation of sapogenin anions typically observed at *m/z* 459.3838 and 475.3787 [47]. As displayed in the MS^2^ spectrum of mal-ginsenoside Rd (Figure 5B), we could attribute the NL of 86.0004 Da (*m/z* 1031.5437 > 945.5421) or 104.0110 Da (*m/z* 1031.5437 > 927.5307) to the elimination of a malonyl substituent ([M−H−Mal]^–^) or ([M−H−Mal−H_2_O]^−^). Other fragments of *m/z* 945.5421 > 783.4895 > 621.4363 > 459.3840 were consistent with the successive eliminations of three Glc groups in m-Rd and the fragments of *m/z* 765.4790 attributed to the elimination of a molecule of H_2_O from the fragments of *m/z* 783.4895. Similar fragmentation patterns were observed in the CID-MS^2^ spectrum of an unknown compound **267**# (*t*_R_ 16.92 min), which gave the deprotonated precursors at *m/z* 1117.5435. Accordingly, we could identify its molecular formula as C_54_H_86_O_24_. In its MS^2^ spectrum, the product ions of *m/z* 945.5402 ([M−H−2Mal]^−^) and *m/z* 927.5345 ([M−H−2Mal−H_2_O]^−^), could testify to the presence of two malonyl groups (Figure 5B). Other fragments, such as *m/z* 783.4887 ([M−H−2Mal−Glc]^−^), 765.4794 ([M−H−2Mal−Glc−H_2_O]^−^), 621.4367 ([M−H−2Mal−2Glc]^−^), and 459.3850 ([PPD−H]^−^), could suggest malonylfloralginsenoside Rd6 or isomer (PPD-3Glc-Dimal).

Eighteen OA-type saponins involving a pentacyclic oleanolic acid sapogenin were characterized from PGL/PQL/PNL (5.26% of the total amount). The single deprotonated precursor ion ([M−H]^−^), and the OA sapogenin ion at *m/z* 455.3525 and the NL 176.0321 Da for GlurA, could assist to rapidly characterize an OA ginsenoside [17]. The CID-MS^2^ fragmentations for the reference compound, ginsenoside Ro, was illustrated (Figure 5C), which displayed the abundant deprotonated precursor at *m/z* 955.4934. Its MS/MS fragmentation could eliminate the connected sugar residues thus generating rich product ions at *m/z* 793.4384 ([M−H−Glc]^−^), *m/z* 731.4378 ([M−H−Glc−CO_2_−H_2_O]^−^) owing to the elimination of one carboxyl group of and one hydroxyl group, and the sapogenin ion at *m/z* 455.3537 ([OA−H]^−^). For the unknown compound **197**# (*t*_R_ 12.92 min, C_53_H_84_O_23_) giving the deprotonated precursors at *m/z* 1087.5323 ([M−H]^−^), its molecular formula was deduced to be C_53_H_84_O_23_. Its CID-MS^2^ spectrum exhibited the product ions of *m/z* 925.4783 ([M−H−Glc]^−^), 793.4323 ([M−H−Glc−Xyl]^−^), 731.4347 ([M−H−Glc−Xyl−CO_2_−H_2_O]^−^, and 455.3517 ([OA−H]^−^) (Figure 5C), which could inform the presence of GlurA, two Glc, and Xyl, attached to the OA sapogenin. By further searching the in-house ginsenoside library, component **197**# was tentatively characterized as stipuleanoside R2 or isomer (OA-GlurA-2Glc-Xyl).

OT-type ginsenosides are a subclass of saponins characteristic of *P. quinquefolius* [11], and we could characterize a total of thirteen ginsenosides of this type accounting for 3.80% of the total amount. The composition of the precursor ion clusters for this OT-type was very similar to the PPT- and PPD-type above mentioned, that is, the concomitant occurrence of the deprotonated precursors and the FA adducts. The sapogenin ion at *m/z* 491.3736 was characteristic of the OT-type ginsenosides [47]. For the reference compound, 24(*R*)-pseudoginsenoside F11 (*t*_R_ 9.81 min; C42H72O14), rich product ions, observed at *m/z* 799.4902 ([M−H]^−^), 653.4265 ([M−H−Rha]^−^), and 491.3704 ([OT−H]^−^), were illustrated (Figure 5D). In the case of an unknown compound **113**# (*t*_R_ 9.44 min) which gave the deprotonated precursors at *m/z* 785.4680 ([M−H]^−^), we could characterize its molecular formula as C_41_H_70_O_14_. Upon the CID-MS/MS fragmentation, rich product ions at *m/z* 653.4273 ([M−H−Rha]^−^), and the characteristic sapogenin ion at *m/z* 491.3726 ([OT−H]^−^) were observed (Figure 5D). After searching the in-house ginsenoside library, compound **113**# was tentatively characterized as pseudoginsenoside Rt2 or isomer (OT-Glc-Xyl).

Aside from these four subtypes, we also characterized the other subtypes of ginsenosides from these three leaf samples, such as the C-17 side-chain varied [9,11]. These ginsenosides possessed multiple variations on the C-17 side chains, such as hydroxylation (**96**#) and C-20 dehydration (**322**#/**317**#/**305**#). A total of 45 C-17 side-chain varied saponins (13.16% of the total amount) and 63 others (18.42% of the total amount) were identified or tentatively characterized. Notably, among the 342 saponins characterized in this work, 81 thereof (23.68%) had unknown masses or potentially new sapogenins (Appendix A). However, to exactly identify the glycosylation sites, the configuration of the glycosidic bonds, the linkage pattern, and those unknown substituents within these saponins, more dimensional structural information is needed, such as to isolate them by phytochemical approaches and exactly identify the structures by NMR and other spectroscopic methods.

The structural features for these saponins characterized from these three different leaf samples were summarized. Firstly, sapogenin or malonylation-oriented retention was observed. As visualized by a 2D scatter plot of *m/z* (200–1500) VS. *t*_R_ (1.63–20.71 min) in Figure 6A, for the PPT- and PPD-types, more sugars contained (with larger *m/z*) could enlarge the polarity and weaken the retention on the RP column, and the PPD-type was generally better retained than the PPT-type. The acidic saponins, including both the OA-type and malonylated, were well retained through elution by the mobile phase consisting of FA as the additive. The C-17 side-chain varied saponins, due to the occurrence of both the polar and non-polar variations, showed wide distribution across the whole elution time. Secondly, the percentage distribution for different subtypes was plotted by the pie chart in Figure 6B. Evidently, the PPD-, PPT-, C-17 side-chain varied, and the malonylated type, occupied very large proportions, aside from those miscellaneous. Thirdly, evaluated from each species, these three different leaf samples shared 51 common saponins, and 101, 55, and 72 saponins were only detected from PGL, PQL, and PNL, respectively (Figure 6C). More reliable differential components among these three leaf samples could be obtained based on the untargeted metabolomics analyses of the multi-batch samples.

### 2.4. Holistic Comparison of the Ginsenosides among PQL, PGL, and PNL by the Untargeted Metabolomics Workflows

On the basis of the systematic chemical elucidation, in the next step, the multiple batches of three leaf samples were compared in a holistic manner using an untargeted metabolomics approach. Multivariate statistical analysis based on the negative MS^E^ data of a total of 33 batches of PGL, PQL, and PNL samples was performed to unveil the potential markers diagnostic for their differentiation [17]. After being processed by the Progenesis QI software, a data table including 13,942 metabolic features was obtained, and these variables were further screened by “80% rule” and “30% variation”, leaving 13,115 ions. The normalized data were used for the pattern recognition chemometrics by PCA (unsupervised) and OPLS-DA (supervised) in turn [38]. In the score plot of PCA (Appendix A), the QC samples were tightly clustered around the center area, which could indicate good quality for the multi-batch data. Three remarkable clusters, representative of PGL, PQL, and PNL got well grouped, which demonstrated their significant chemical differences. The values of R^2^X (cum) 0.920 and Q^2^ (cum) 0.803 could indicate the PCA model showed good fitness and predictability. Then, the supervised OPLS-DA was utilized to discover the differential components among three leaf samples. The fitted OPLS-DA classifier exhibited good fitness (*R*^2^X 0.707, *R*^2^Y 0.955) and predictability (*Q*^2^ 0.948), in which each group was remarkably separated from the other (Figure 7A). The permutation test could indicate this OPLS-DA model was not overfitted (Figure 7C). Variable importance in the projection (VIP) can unveil the importance of each variable to the observed clustering. A total of 1243 differential ions showed VIP > 1.0, and when the VIP cutoff was set at 5.0, 72 major differential ions were obtained (Figure 7B). A heat map was plotted showing the abundance difference for these 72 differential ions among the tested leaf samples, as shown in Figure 7D.

By consulting the multi-component characterization results (Appendix A), these 72 differential ions could be assigned to 42 ginsenoside compounds (**M1** to **M42**), as given in Appendix A. Notably, 20 potential marker ginsenosides were confirmatively identified by comparison with the reference compounds. We had plotted the abundance differences for these 42 markers in the box charts (Figure 8), and important identification points for differentiating among PGL, PQL, and PNL were summarized as follows.

(i) Some saponins were rich in one of these three varieties, and could be used as their identification markers: F1 (**M4**), F3 (**M5**), 20(*S*)-Rh1-6’-acetate/isomer (**M7**), F5 (**M8**), vina-R4 (**M11**), p-RC1/isomer (**M13**), isomer of F5 (**M20**), floral-P/isomer (**M38**), and M7cd/isomer (**M39**) in PGL; 24(*R*)-p-F11 (**M1**), q-L9 (**M23**), and pseudo-Rt5 (24*R*) (**M41**) in PQL; Ra1 (**M9**), noto-Fa (**M12**), Ra2 (**M17**), noto-Rt or isomer (**M31**), and I/isomer (**M33**) in PNL;

(ii) PGL and PQL showed similar saponin composition (Figure 4 and Figure 7A), with richer Re (**M2**), 6″-*O*-ace-Rg3 or its isomers (**M10**, **M15** and **M26**), Rg2 (**M16**), m-Rb1 (**M27**), q-L2 (**M30**), noto-Fp1 (**M37**), and Rd (**M42**), than PNL; PGL and PNL contained richer Rc (**M21**) and Rb1/isomer (**M35**), compared with PQL; the contents of and Rb3 (**M3**), m-Rc isomer (**M6**), and floral-P or isomer (**M28** and **M40**), were larger than PNL.

## 3. Materials and Methods

### 3.1. Chemicals and Materials

A total of 60 ginsenoside compounds (structures in Appendix A and detained information in Appendix A), were purchased from Chengdu Desite Biotechnology Co., Ltd. (Chengdu, China), Shanghai Standard Biotech. Co., Ltd. (Shanghai, China), or isolated from the root of *P. ginseng* by the authors, respectively. Acetonitrile, methanol (Fisher, Fair lawn, NJ, USA), and formic acid (FA) (ACS, Wilmington, NC, USA), were all of the LC-MS grade. Ultra-pure water (18.2 MΩ cm at 25 °C) was prepared utilizing a Milli-Q Integral 5 water purification system (Millipore, Bedford, MA, USA). Related information with regard to multiple batches of three *Panax*-derived ginseng samples was provided in Appendix A. Voucher specimens of these analytes were deposited at the authors’ laboratory in the State Key Laboratory of Component-based Chinese Medicine, Tianjin University of Traditional Chinese Medicine (Tianjin, China).

### 3.2. Sample Preparation

A QC sample (QC1) was prepared for the LC-MS method development by pooling an equal volume of three representative samples (PNL-1, PQL-1, and PGL-1), whilst another QC sample (QC2) was prepared by mixing 33 batches of the leaf samples in equal volume to monitor the stability of the system all through the analytical batch. The detailed process of sample preparation was as follows: 100 mg of accurately weighed fine powder (<40 mesh) of each sample was dissolved in 10 mL of 70% (*v*/*v*) methanol and then extracted in a water bath (40 °C) with ultrasound assistance (400 W, 40 kHz) for 1 h [13]. The extract was next centrifuged at 3219× *g* (4000 revolutions per minute, rpm) for 10 min, with the supernatants transferred into a 10 mL-volumetric flask. The extraction process was repeated, and the pooled extract was subsequently centrifuged at 11,481× *g* (14,000 rpm) for 10 min. Ultimately, the supernatant was used as the test solution (10 mg/mL).

### 3.3. UHPLC/IM-QTOF-MS

Profiling and characterization of the multicomponents of PGL, PQL, and PNL were performed on an ACQUITY UPLC I-Class/Vion^TM^ IM-QTOF system (Waters, Milford, MA, USA). A BEH Shield RP18 column (2.1 × 100 mm, 1.7 µm), hyphenated with a VanGuard Pre-column (Waters; 2.1 × 50 mm, 1.7 µm) kept at 30 °C, was utilized for chromatographic separation, which was eluted by a binary mobile phase containing 0.1% formic acid in water (A) and acetonitrile (B). An optimal gradient elution program was utilized as follows: 0−4 min, 15−23% (B); 4–8 min, 23–28.5% (B); 8–10 min, 28.5–33% (B); 10–13 min, 33% (B); 13–16 min, 33–34% (B); 16–19 min, 34–95% (B); and 19–21 min, 95% (B). The flow rate was set at 0.3 mL/min and the injection volume was 3 µL.

High-resolution MS spectra were acquired by a hybrid HDMS^E^-MS/MS (HDMS^E^-HDDDA) experiment on the Vion^TM^ IM-QTOF mass spectrometer operating in negative mode (Waters). The ion source parameters were set as the following: capillary voltage, −1.0 kV; cone voltage, 20 V; source offset, 80 V; source temperature, 120 °C; desolvation gas temperature, 500 °C; desolvation gas flow (N_2_), 800 L/h; and cone gas flow (N_2_), 50 L/h. Default parameters were defined and CCS calibration was performed using a mixture of calibrators according to the manufacturer’s guidelines for the traveling wave IMS separation [49]. The QTOF analyzer scanned over *m/z* 200–1500 at 0.3 s per scan and 6 eV was set for low energy to record the information of all precursors by full scan. RCE of 80–100 eV in HDMS^E^ and MDRCE of 30 eV in low mass ramp and 70 eV in high mass ramp in HDDDA, were set. The HDMS^E^-MS/MS experiment was composed by setting HDMS^E^ and HDDDA. In HDMS^E^, the ramp collision energy (RCE) of 80–100 eV was set to record the MS^2^ data by CID, with the scan rate at 0.3 s/scan. In HDDDA, the Top N function was utilized: (1) when the intensity exceeded 500 detector counts, MS^2^ fragmentations of three most intense precursors were automatedly triggered; and (2) the MS^2^ acquisition stopped if either TIC (total ion current chromatogram) exceeded 100 detector counts or time exceeded 0.4 s. Mass-dependent ramp collision energy (MDRCE) in low mass ramp and in high mass ramp were set at 30 eV and 70 eV in HDDDA. MS data acquisition was controlled by the UNIFI 1.9.3.0 software (Waters Corporation). Calibration of the data was conducted by constantly infusing 200 ng/mL of leucine enkephalin solution (Sigma-Aldrich, St. Louis, MO, USA) at a flow rate of 10 µL/min.

### 3.4. Performance Comparison between HDMS^E^-HDDDA and Three other MS^2^ Data Acquisition Methods

The developed hybrid scan method was compared with DDA, DDA containing precursor ions list (DDA-PIL), and HDMS^E^ available on the Vion IM-QTOF mass spectrometer, with the same modes and the same parameters set to make all the results comparable. The information for the PIL was included in Appendix A. The raw data were all processed by UNIFI for data correction, peak extraction, and peak annotation [17].

### 3.5. Automated Peak Annotation Workflows Facilitated by UNIFI^TM^ and an in-House Ginsenoside Library

UNIFI 1.9.3.0 platform driven by the in-house ginsenoside library (containing 573 ginsenosides) was established to process data. To achieve the automatic peak annotation, incorporating information about these saponins into the UNIFI platform to match those obtained MS^1^ and MS^2^ data with the theoretical values. “Identified Components” with detailed information (retention time, aligned MS^2^ fragments, CCS, etc.) were listed according to the predefined matching rules. Those resolved peaks with no hits in the in-house library were assigned as “Unknown Components”. Adduct ions filtering and MS^2^ data analysis were further conducted to re-confirm these identifications, by removing the false positive results [33].

Uncorrected data were corrected by the Lock Mass at *m/z* 554.2620 (ESI^−^), with a combined width of 3 scans and a mass window of 0.5 *m/z*. High-energy and low-energy intensity thresholds were separately set at 100.0 counts and 500.0 counts. Tolerance for both target match and fragment match was 10 ppm. To remove the redundant adduction species, multiple adduct ions, containing [M–H]^−^, [M−H+CH_3_COOH]^−^, [M−H+HCOOH]^−^, and [M+Cl]^−^, were used.

### 3.6. Holistic Ginsenoside Comparison among PGL, PQL, and PNL Based on Untargeted Metabolomics

Untargeted metabolomics differential analysis for the ginsenosides from the leaves of three *Panax* species (*P. ginseng*, *P. quinquefolius*, and *P. notoginseng*) was performed by analyzing the negative MS^E^ data of 33 batches of samples. Initially, the multi-batch raw data were loaded into UNIFI for the first step of data correction by reference to *m/z* 554.2620 for the lock mass data. The corrected data were further processed by the Progenesis QI 2.4.6 software (Waters Corporation). Isotope and adduct fusion were utilized to reduce the number of detected metabolic features. The adduct forms, involving [M−H]^−^, [M−H+HCOOH]^−^, [M+Cl]^−^, [M−2H]^2−^, [M−2H+HCOOH]^−^, [2M−H]^2−^, and [M−2H+2HCOOH]^2−^, in the negative mode were selected or in-house edited [17]. A data matrix, including the information of *t*_R_, *m/z*, and normalized abundance, was obtained. The “30% variation” and “80% rule” rules were utilized to filter these variables [50]. The selected variables were loaded into the SIMCA-P 14.1 software (Umetrics, Umea, Sweden) for multivariate statistical analysis by PCA and OPLS-DA. Those variables showing VIP > 5.0 were selected as the potential markers for differentiating among PGL, PQL, and PNL [24].

## 4. Conclusions

For the comprehensive ginsenosides characterization and the holistic comparison among PGL, PQL, and PNL, an integral strategy, by combining enhanced MS^2^ acquisition, intelligent peak annotation, and chemometrics, was presented in the current work. A novel hybrid scan approach (HDMS^E^-PIL/HDDDA), enabling the IM separation of all precursor ions and the alternate DIA/DDA acquisitions, was established by UHPLC/IM-QTOF-MS in the negative ESI mode. In total, 342 ginsenosides were identified or tentatively characterized from PGL, PQL, and PNL, by feat of the intelligent data processing workflows facilitated by the UNIFI^TM^ platform and searching against an in-house ginsenoside library. The holistic comparison among the 33 batches of leaf samples from three *Panax* species could unveil 42 potential ginsenoside markers. In particular, some characteristic ginsenosides showed the potential for the precise authentication among PGL, PQL, and PNL. The hybrid scan approach could balance between the coverage on the interested components and the spectral reliability, generate high-definition MS^1^ and MS^2^ spectra, and provide CCS information having the potential to discriminate isomers. It is the first report that systematically compares the metabolome differences among the leaves of the three most important *Panax* species, and the results obtained can greatly benefit the quality control of the natural products derived from the *Panax* genus.

## Figures and Tables

**Figure 1 molecules-27-05549-f001:**
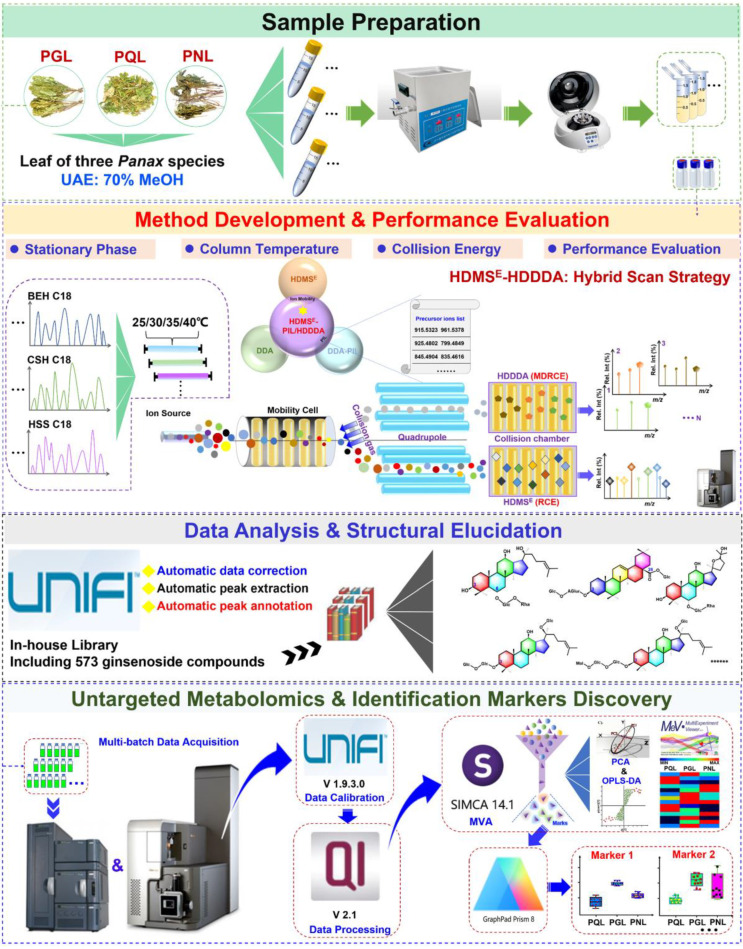
A general technical roadmap of the proposed analytical strategy, by integrating a hybrid scan approach (HDMS^E^-HDDDA), intelligent data processing workflows facilitated by UNIFI and an in-house ginsenoside library, and untargeted metabolomics, for the holistic comparison of ginsenosides from the leaves of three *Panax* species (*P. ginseng*, *P. quinquefolius*, and *P. notoginseng*).

**Figure 2 molecules-27-05549-f002:**
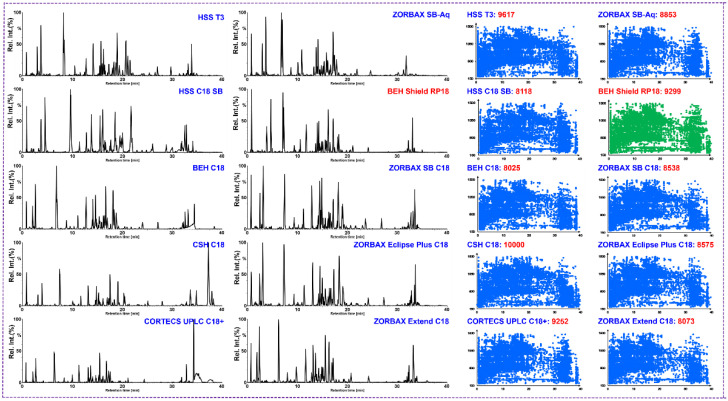
Method development for UHPLC/IM-QTOF-MS by evaluating the separation of ginsenosides on ten candidate UHPLC columns (left: the base peak chromatograms obtained on ten candidate columns; right: the two-dimensional scatter plots of the resolved peaks based on the *t*_R_ and *m/z* information).

**Figure 3 molecules-27-05549-f003:**
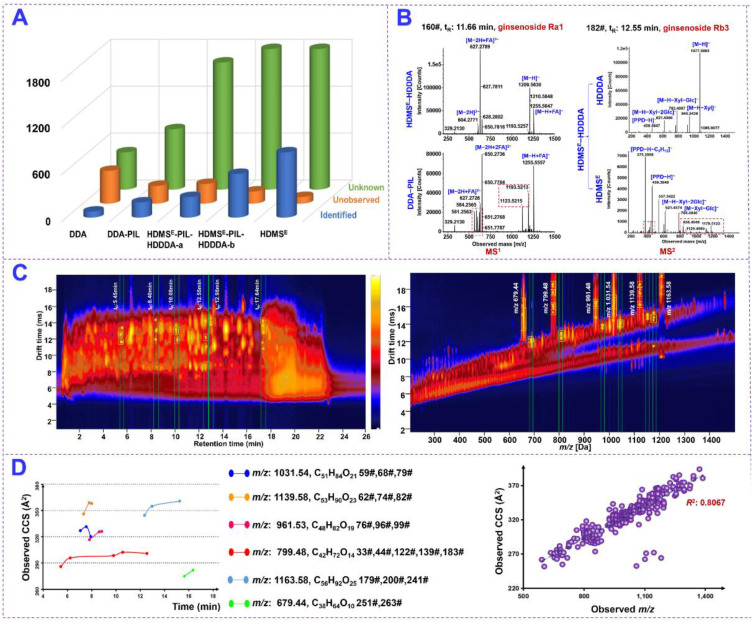
Superiority demonstration for the developed hybrid scan approach of HDMS^E^-PIL/HDDDA in the characterization of ginsenosides from PGL, PQL, and PNL. (**A**): A three-dimensional histogram showing the numbers of “Identified Components”, “Unobserved Components”, and “Unknown Components” acquired from the QC1 sample by four different scan approaches; (**B**): Comparing the MS^1^ spectra acquired by the hybrid scan approach and DDA-PIL (showing the better resolution facilitated by enabling IM separation) and the MS^2^ spectra obtained by HDMS^E^ and HDDDA of the hybrid scan approach (showing better spectra quality for DDA); (**C**): 2D heat maps of drift time VS *t*_R_ (showing the separation for 6 groups of co-eluting components) and drift time VS *m/z* (showing the separation for 6 groups of isomers); (**D**): the CCS distribution for 6 groups of isomers and the 2D scatter plot (CCS VS *m/z*) for the 342 components characterized from PGL, PQL, and PNL.

**Figure 4 molecules-27-05549-f004:**
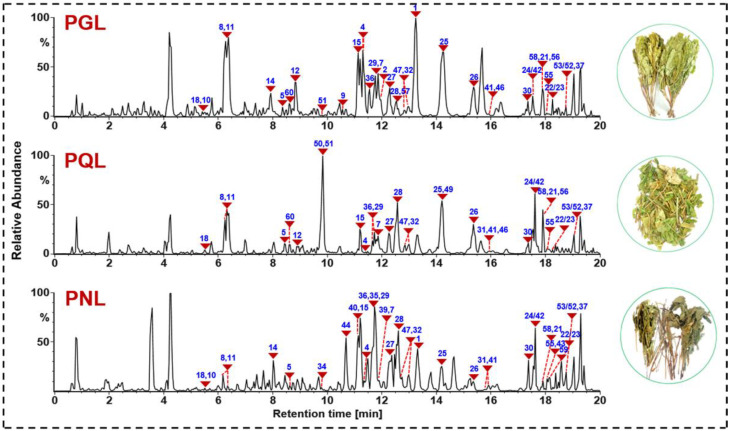
Base peak chromatograms for the representative samples of PGL (PGL-1), PQL (PQL-1), and PNL (PNL-1). The peaks identified by comparison with the reference compounds are annotated with their numbering in Appendix A.

**Figure 5 molecules-27-05549-f005:**
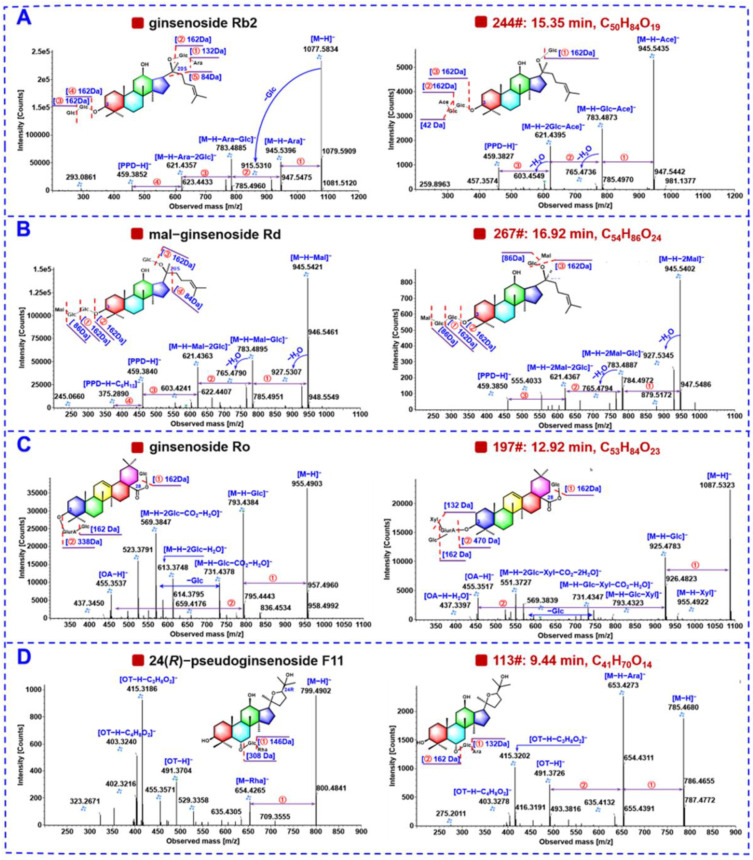
Illustration for characterizing the PPD- (**A**), malonylated (**B**), OA- (**C**), and OT-(**D**) type ginsenosides, from PGL/PQL/PNL using the representative compounds.

**Figure 6 molecules-27-05549-f006:**
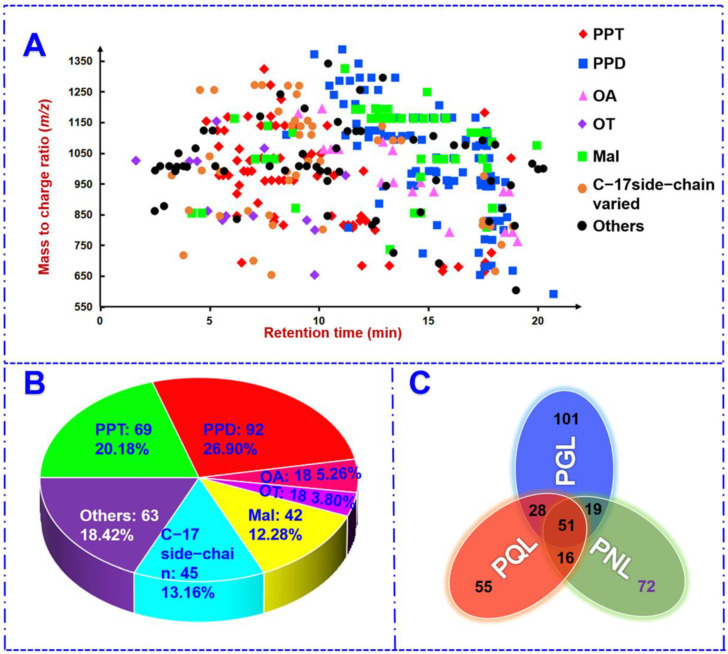
A global summary of the structural features of 342 ginsenosides characterized by PGL, PQL, and PNL. (**A**): A 2D scatter plot of all ginsenoside structures by *t*_R_ and *m*/*z*; (**B**): a pie chart sorted by the subcategory of ginsenosides; (**C**): a Venn diagram.

**Figure 7 molecules-27-05549-f007:**
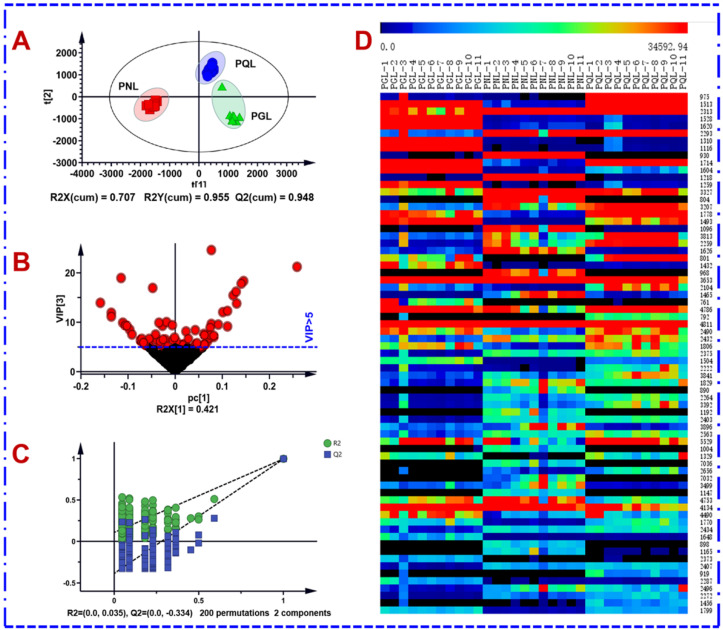
Multi-variate statistical analysis of 33 batches of the leaf samples of three *Panax* species (PGL/PQL/PNL) using OPLS-DA based on the negative-mode MS^E^ data. (**A**): Score plot; (**B**): VIP plot; (**C**): permutation test; (**D**): heat map of 72 differential ions among the tested leaf samples.

**Figure 8 molecules-27-05549-f008:**
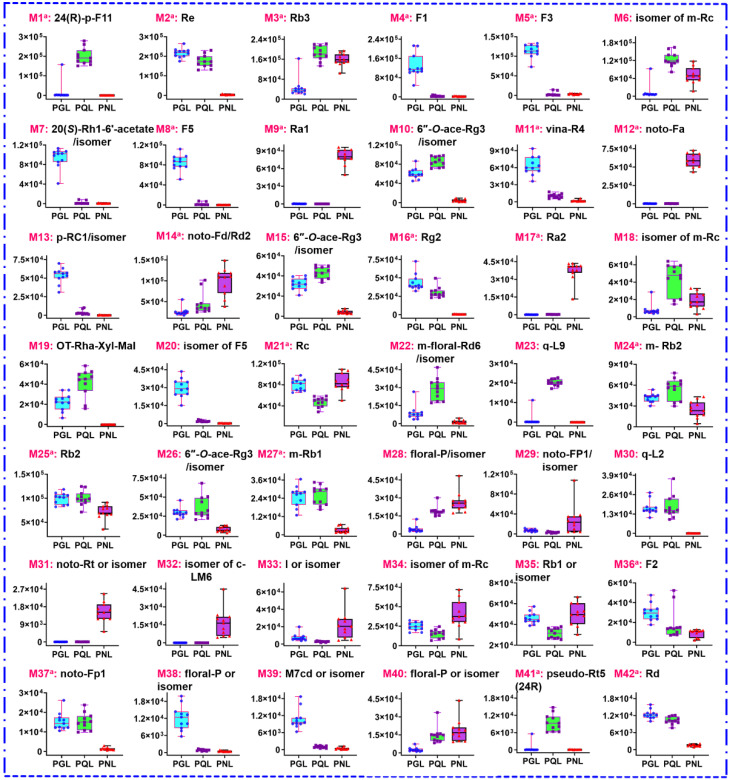
Box charts illustrating the content difference of 42 important ginsenoside markers among PGL, PQL, and PNL. (c-: chikusetsusaponin; q-: quinquenoside; noto-: notoginsenoside; floral-: floralginsenoside; p-: pesudoginsenoside; vina-: vinaginsenoside; m-: malonylginsenoside; ace-: acetyl; m-floral-: malonylfloral).

## Data Availability

The data presented in this study are available in the article and Appendix A.

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
