# Peer review of "Integrating Enhanced Profiling and Chemometrics to Unveil the Potential Markers for Differentiating among the Leaves of Panax ginseng, P. quinquefolius, and P. notoginseng by Ultra-High Performance Liquid Chromatography/Ion Mobility-Quadrupole Time-of-Flight Mass Spectrometry"

_molecules, 2022, doi:10.3390/molecules27175549_

Round 1
Reviewer 1 Report
The manuscript “Integrating Enhanced Profiling and Chemometrics to Unveil the 2 Potential Markers for Differentiating Among the Leaves of 3 Panax ginseng, P. quinquefolius and P. notoginseng by Ultra-high 4 Performance Liquid Chromatography/Ion Mobility-Quadrupole 5 Time-of-flight Mass Spectrometry” reports on a comparison of extracts from leaves of three different strains of ginseng utilizing a hybrid HPLC-MS scan approach (HDMSE-HDDDA).
The manuscript is well written, the methodology is systematically well thought out and highly suited to the problem set, and the presentation of the results is of high quality. The comparison of different C18 columns and column temperatures increases confidence in the transfer of the method to different experimental setups. The addition of ion mobility data to the MS study helps to validate the results and allows to assignment of isomers that otherwise would not have been identified individually. The MSMS characterization of the PPD-, malonylated, OA-, and OT-type 367 ginsenosides is compelling and the summary of important ginsenoside markers allows for a quick comparison of the results for the different strains. The SI provides a good summary of compounds, chemical formulas and masses that can be used for setting up similar experiments. Often provided UV/Vis data is missing, but not necessary given the compelling analysis, the lower sensitivity of UV/Vis detectors and the ambiguity in assigning bands based on this type of data.
Author Response
Reply: It's a great honor for our manuscript to get the reviewer's approval. We are very grateful to the reviewer for the objective comments, careful summary, and the proposing of good suggestions. We agree with the reviewer that it’s better to provide the UV information for the identified components. Actually, most of these characterized components are of low content, which fail to give definite UV signal.
Again, we thank the reviewer for the efficient and careful review work.
Reviewer 2 Report
Panax species are widely used in Chinese medicine and as food additives. Detailed characterization of biologically active saponins is very important to ensure quality of the products. Moreover this information allows differentiation of members of Panax genus on the basis of cheminformatics. The authors used an original approach involved ion mobility and high resolution mass spectrometry and succeeded in identification or characterization of a number of saponins. This data was successfully applied for differentiation of three species of Panax genus. For these point I consider the manuscript worthy of publication. But some improvements should be made to support it:
1. Some information concerning resolution and mass accuracy of the instrument should be added. The authors applied HRMS but often indicates only two digits after zero for m/z ratios.
2. Please add some discussion concerning elimination of water from the molecules on the analytes in course of MS/MS experiments. For example, there are intense peaks corresponding to this process in case of malonylginsenosides, but MS/MS mass spectra of ginsenosides Rb2 and #244, having hydroxyl group in the same positions, do not contain such ions.
3. Please add some discussions concerning formation of [M−H−(X)*Sugar Residue-CO2-H2O]− ions (for example m/z 731.4378 for ginsenosides Ro.
Author Response
Panax species are widely used in Chinese medicine and as food additives. Detailed characterization of biologically active saponins is very important to ensure quality of the products. Moreover this information allows differentiation of members of Panax genus on the basis of cheminformatics. The authors used an original approach involved ion mobility and high resolution mass spectrometry and succeeded in identification or characterization of a number of saponins. This data was successfully applied for differentiation of three species of Panax genus. For these point I consider the manuscript worthy of publication. But some improvements should be made to support it.
Reply: We greatly appreciate the reviewer for the efficient review work and the proposing of constructive suggestions to improve the quality of our manuscript. We agree to the following valuable modifications and make corresponding amendments. Our response to the comments is appended item by item as follows.
General points:
- Some information concerning resolution and mass accuracy of the instrument should be added. The authors applied HRMS but often indicates only two digits after zero for m/z
Reply: We thank the reviewer for this comment. For the Vion IM-QTOF instrument we utilized in the current work, the resolution of the TOF analyzer was not a constant (different from the Q-Orbitrap series), which typically reached 25,000-40,000 at the calibration stage. So we do not add this information in the Experimental section.
To the accurate MS data, we follow this suggestion, and change them with four digits after zero, in the revised version.
- Please add some discussion concerning elimination of water from the molecules on the analytes in course of MS/MS experiments. For example, there are intense peaks corresponding to this process in case of malonylginsenosides, but MS/MS mass spectra of ginsenosides Rb2 and #244, having hydroxyl group in the same positions, do not contain such ions.
Reply: We thank the reviewer for this comment. In the negative ion mode, the neutral loss of H2O is a common form observed in the MS2 spectra of ginsenosides (such as the fragments at m/z 765.4736 and 603.4549 in the MS/MS spectrum of #244). The neutral losses of a malonyl group by 86.00 Da plus a molecule of H2O by 104.06 Da or even the dimalonyl groups by172.00 Da plus a molecule of H2O by190.06 Da, were the characteristic fragmentation features for malonyl substituted ginsenosides. We have adequately added the discussion in the text of the revised manuscript.
- Please add some discussions concerning formation of [M−H−(X)*Sugar Residue-CO2-H2O]− ions (for example m/z 731.4378 for ginsenosides Ro).
Reply: We thank the reviewer for this suggestion. We have added the related discussion in the text of the revised version, to give a more comprehensive description of the negative CID-MS2 fragmentation features about the OA-type saponins.
Reviewer 3 Report
The manuscript is interesting, and this study is necessary, considering the importance of employed new and efficient protocols to metabolome difference among the leaves of Panax species, using an ultra-high performance liquid chromatography/ion mobility-quadrupole time-of-flight mass spectrometry (UHPLC/IM-QTOF-MS), which showed the potential to discriminating isomers. The results can benefit the quality control of the natural products derived from the Panax genus. It is understandable, and it is well organized. I find no problems with the scientific approach or technical content presented by the authors in this manuscript.
Author Response
Reply: We are greatly honored to have your affirmation and thank you very much for your systematic summary and endorsement of the manuscript. In the future work, we will continue to work hard with a earnest and rigorous attitude.
